# Protein determinants of dissemination and host specificity of metallo-β-lactamases

Carolina López [1], Juan A. Ayala [2], Robert A. Bonomo[3,4,5], Lisandro J. González [1,6] & Alejandro J. Vila [1,5,6]

The worldwide dissemination of metallo-β-lactamases (MBLs), mediating resistance to carbapenem antibiotics, is a major public health problem. The extent of dissemination of MBLs such as VIM-2, SPM-1 and NDM among Gram-negative pathogens cannot be explained solely based on the associated mobile genetic elements or the resistance phenotype. Here, we report that MBL host range is determined by the impact of MBL expression on bacterial fitness. The signal peptide sequence of MBLs dictates their adaptability to each host. In uncommon hosts, inefficient processing of MBLs leads to accumulation of toxic intermediates that compromises bacterial growth. This fitness cost explains the exclusion of VIM-2 and SPM-1 from *Escherichia coli* and *Acinetobacter baumannii*, and their confinement to *Pseudomonas aeruginosa*. By contrast, NDMs are expressed without any apparent fitness cost in different bacteria, and are secreted into outer membrane vesicles. We propose that the successful dissemination and adaptation of MBLs to different bacterial hosts depend on protein determinants that enable host adaptability and carbapenem resistance.

[1] Instituto de Biología Molecular y Celular de Rosario (IBR, CONICET-UNR), S2000EZP Rosario, Argentina. [2] Centro de Biología Molecular Severo Ochoa, Agencia Estatal Consejo Superior de Investigaciones Científicas (CSIC-UAM), Campus de Cantoblanco, 28049 Madrid, Spain. [3] Medical Service, Louis Stokes Cleveland Department of Veterans Affairs Medical Center, Cleveland, OH 44106, USA. [4] Departments of Medicine, Pharmacology, Molecular Biology and Microbiology, Biochemistry, Proteomics and Bioinformatics, Case Western Reserve University School of Medicine, Cleveland, OH 44106, USA. [5] CWRU-Cleveland VAMC Center for Antimicrobial Resistance and Epidemiology (Case VA CARES), Cleveland, OH 44106, USA. [6] Área Biofísica, Facultad de Ciencias Bioquímicas y Farmacéuticas, Universidad Nacional de Rosario, S2002LRK Rosario, Argentina. Correspondence and requests for materials should be addressed to L.J.G. (email: lgonzalez@ibr-conicet.gov.ar) or to A.J.V. (email: vila@ibr-conicet.gov.ar)

Carbapenems are "last-resort" β-lactam antibiotics[1]. Unfortunately, the worldwide spread of carbapenemases (carbapenem-inactivating enzymes) severely compromises our therapeutic arsenal to fight life-threatening bacterial infections and represents a major challenge for public health. Owing to this scenario, the World Health Organization (WHO) has identified carbapenem-resistant *Acinetobacter baumannii*, *Pseudomonas aeruginosa*, and *Enterobacteriaceae* as Priority 1 ("critical") microorganisms for the development of new drugs[2]. Metallo-β-lactamases (MBLs) are Zn(II)-dependent enzymes that constitute one of the major groups of carbapenemases[3–6]. Despite active research on this field, inhibitors of MBLs are not commercially available and few β-lactams are in development that are stable to hydrolysis[7,8].

The most relevant MBLs in term of clinical relevance and geographic dissemination are the families of plasmid-encoded NDM, VIM, SPM, and IMP. VIM-type enzymes (Verona integron-encoded MBLs) were first identified in 1996, now encompassing 60 different allelic variants. VIM-2 is the most commonly reported MBL worldwide, present in outbreaks in *P. aeruginosa*[9], being rarely found in *Enterobacteriaceae*. The São Paulo MBL (SPM-1) is unique since it is represented by only one allele that is host-specific[10], being confined to *P. aeruginosa*[11]. Instead, the New-Delhi Metallo-β-lactamase (NDM) has been identified in strains of *Enterobacteriaceae*, *P. aeruginosa*, *A. baumannii*, and other Gram-negative non-fermenters, showing the fastest and widest geographical spread among MBLs since its first identification in 2008[12,13]. These MBLs are present in all microorganisms labeled as "Priority 1", but their preferential distribution into different bacterial hosts is not understood.

Among Gram-negative bacteria, MBLs are synthesized as cytoplasmic precursors carrying an N-terminal signal sequence (the signal peptide), which directs them to the secretion machinery for their translocation into the periplasmic space[14,15]. VIM, IMP, and SPM are soluble periplasmic proteins[4]. In contrast, NDM-like β-lactamases are lipoproteins anchored to the outer membrane in Gram-negative bacteria[16–19]. This unique cellular localization favors secretion of NDM variants in outer membrane vesicles (OMVs)[17], as it has been shown in laboratory strains of *E. coli* and for *bla*~NDM-1~-harboring clinical isolates of *Enterobacter cloacae*[17]. Protein transport by vesicles is advanced as a mechanism favoring the dissemination of NDM-1, since NDM-containing vesicles can protect bacteria otherwise susceptible to β-lactams, thus enhancing the communication between antibiotic-resistant and susceptible bacteria[17]. This communication allows for horizontal gene transfer to occur between bacteria by conjugation, transformation, transduction, or OMV-mediated gene transfer[20,21]. Whether this vesicle-mediated protein transport is a general mechanism for all MBLs and for different organisms is still unknown. On the other hand, bacterial OMVs can also serve as a pathway to release molecules to alleviate envelope stress[22–24]. Thus, we advance the notion that NDM is secreted into OMVs as a mechanism to relieve the periplasmic stress caused by membrane insertion of this protein.

Here, we explore the role of the cellular localization, protein features, and organismal dependence in the secretion of different MBLs in OMVs from three bacterial species. We discover that NDM β-lactamases are uniquely tailored to be exported into OMVs from different Gram-negative pathogens. Instead, VIM-2 and SPM-1 are only secreted into OMVs upon accumulation of precursor species in non-frequent bacterial hosts, as a response to the stress caused by these toxic precursors. This phenomenon accounts for the restriction of these MBLs to *Pseudomonas aeruginosa*, whose fitness is not compromised by expression of these enzymes. Finally, we identify that the peptide leader sequence of MBLs is crucial not only in defining their cellular localization, but also the bacterial host specificity. We propose that the worldwide dissemination of NDM among different bacterial hosts compared to other MBLs is not due to the specific genetic environment of the *bla*~NDM~ gene neither to its resistance phenotype, but to the unique and singular features of this protein. NDM β-lactamases have been shaped during evolution by being able to override the fitness cost and acquire and maintain resistance determinants in the absence of evolutionary pressure.

## Results

**Sorting of MBLs into OMVs is host and protein-dependent.** In order to study the sorting of clinically relevant MBLs in OMVs secreted by different Gram-negative bacteria, we selected NDM-1, VIM-2, and SPM-1. These enzymes belong to three representatives of MBLs with different host distribution and prevalence. As testing organisms, we selected *A. baumannii*, *P. aeruginosa* and *E. coli* (as representative of *Enterobacteriaceae*), i.e., the three pathogens highlighted as "critical" by the WHO[2]. We expressed the *bla*~NDM-1~, *bla*~VIM-2~, and *bla*~SPM-1~ genes (each one with its native signal peptide), under the control of a pTac promoter in the three different genera of bacteria. The expression level of each MBL can be modulated by the concentration of added isopropyl-β-D-thiogalactopyranoside (IPTG). We employed IPTG concentrations that induce expression levels similar to those found in clinical strains[17]. The three MBLs were expressed fused to a C-terminal Strep-tag (ST), to allow homogeneous immunodetection and quantitation both in bacterial cells and in the vesicles. This tag does not interfere with the proper expression, secretion, folding, and activity levels of MBLs[17].

We measured the minimum inhibitory concentrations (MICs) of imipenem, ceftazidime, cefepime, and piperacillin, against the three strains expressing the different MBLs. All were able to confer resistance to these hosts against all tested antibiotics (Supplementary Fig. 1 and Supplementary Table 1). Specifically, NDM-1 conferred the highest resistance levels in *P. aeruginosa* and *A. baumannii*, while SPM-1 had the best performance among the three proteins in *E. coli*. The observed phenotypes do not correlate with the host preferences of VIM-2 and SPM-1. Socha et al.[25] have recently shown that the phenotypic variation in resistance among MBLs in different organisms is explained from a combination of the catalytic efficiency and abundance of functional periplasmic enzyme, in line with previous published observations[26].

Figure 1a (top panel) shows the protein levels of the three MBLs expressed in *E. coli*, *P. aeruginosa* and *A. baumannii*, detected in whole cells. In the case of NDM-1, only the mature protein was detected in the three strains, displaying lower expression levels in *A. baumannii* compared to *E. coli* or *P. aeruginosa*. Instead, VIM-2 and SPM-1 showed accumulation of both the precursor (indicated in red arrows, Fig. 1a) and the mature forms when these proteins were expressed in *E. coli*. Cell fractionation and proteinase K treatment of *E. coli* spheroplasts showed that the precursor forms of VIM-2 and SPM-1 are associated with the periplasmic face of the inner membrane as part of insoluble aggregates (Fig. 1b and Supplementary Fig. 2). In the case of *A. baumannii*, we observed that the synthesis of VIM-2 is highly compromised, detecting only the mature form in low levels. However, at longer times of induction (stationary phase), we detected higher levels of VIM-2 in both precursor and mature forms (Supplementary Fig. 3). On the contrary, the protein levels of SPM-1 expressed from *A. baumannii* cells were high, showing the accumulation of mature and precursor forms. Instead, accumulation of the precursor was not observed in *P. aeruginosa* strains expressing VIM-2 or SPM-1, where only the mature forms of these enzymes were detected (Fig. 1a, top panel).

We purified OMVs secreted from these bacteria, expressing NDM-1, VIM-2, or SPM-1. Transmission electron microscopy

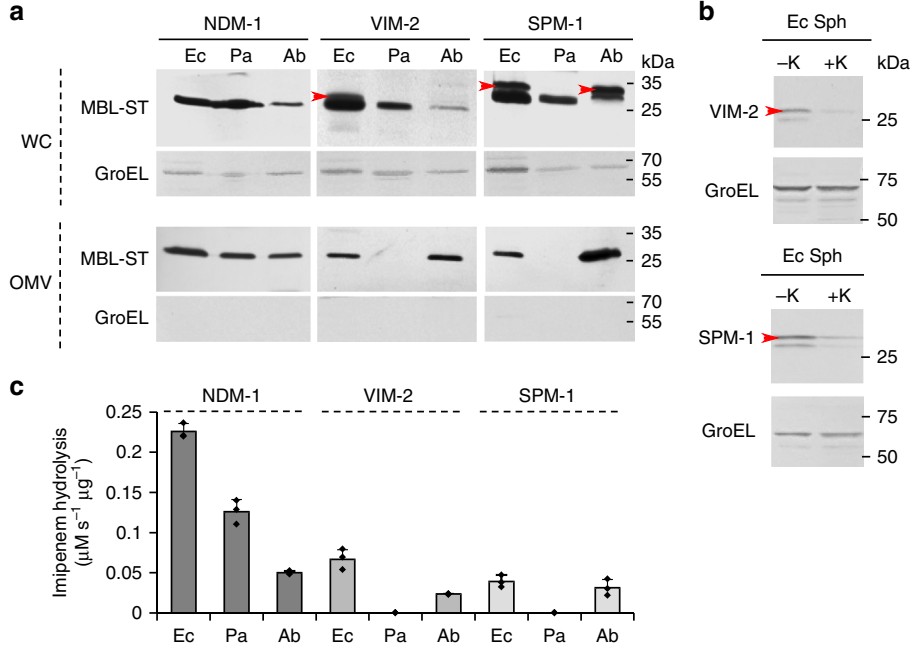

**Fig. 1** MBL sorting in OMVs from *E. coli*, *P. aeruginosa* and *A. baumannii*. **a** NDM-1, VIM-2, and SPM-1 levels (fused to a C-terminal Strep-tag sequence, -ST) by immunoblotting, using anti-Strep-tag antibodies, in whole cells (WC, top panel) and in OMVs purified from the corresponding cell culture supernatants (OMVs, bottom panel) from *E. coli* DH5α [Ec], *P. aeruginosa* PAO1 [Pa] and *A. baumannii* ATCC 17978 [Ab] strains expressing each MBL, after induction with 20 μM IPTG. Gel lanes were loaded with equal amounts of total protein. Red arrows indicate the precursor forms in whole cells from *E. coli*-expressing VIM-2 or SPM-1 and from *A. baumannii*-expressing SPM-1. The presence of cytoplasmic contaminants in vesicle preparations was assessed by immunoblot of GroEL. **b** Immunoblot of spheroplasts from *E. coli*-expressing VIM-2 or SPM-1 treated with ( + K) or without (-K) Proteinase K to asses accessibility of precursor forms. Panels **a** and **b** are representative of two independent experiments. **c** Imipenem hydrolysis rate by OMVs purified from *E. coli*, *P. aeruginosa*, and *A. baumannii* cells expressing NDM-1, VIM-2, and SPM-1. Data correspond to three independent experiments and are shown as the mean value. Error bars represent standard deviations (s.d). Source data are provided as a Source Data file

(Supplementary Fig. 4a and Supplementary Fig. 4b) showed the presence of homogeneous and spherical vesicles (diameters ranging from 20 to 120 nm) free of contaminants, such as fimbriae and flagella. Sodium dodecyl sulfate polyacrylamide gel electrophoresis (SDS-PAGE) demonstrated that MBL expression does not affect the overall protein profiles in OMVs, regardless the MBL cargo (Supplementary Fig. 5). Moreover, these MBLs represent a minor portion of the total protein content of vesicles (Supplementary Fig. 5). NDM-1 levels in OMVs from all three strains were similar (Fig. 1a, bottom panel). Instead, VIM-2 and SPM-1 were found in vesicles from *E. coli* and *A. baumannii*, while traces of these proteins were not found in OMVs purified from *P. aeruginosa* cells expressing these MBLs (Fig. 1a, bottom panel).

We next measured carbapenemase activity in vesicles from all organisms using imipenem as a substrate (Fig. 1c), revealing that the OMVs contain folded and metal-loaded MBLs. However, we stress that the measured activities do not always correlate with the protein levels in vesicles (Fig. 1a, bottom panel).

These observations indicate that MBL secretion into OMVs depends on the properties and characteristics of the protein and the bacterial host. The case of NDM-1 is unique, being secreted as a folded and active protein into OMVs in the three studied organisms that serve as NDM-1 hosts. In contrast, VIM-2 and SPM-1 are not present in vesicles produced by *P. aeruginosa*, the frequent host of these MBLs in clinical isolates, while secretion of these two MBLs into vesicles in *E. coli* and *A. baumannii* correlates with an accumulation of the precursor protein.

**Expression of MBLs in non-frequent hosts compromises fitness.** The production of OMVs was increased (up to 7-fold) in *E. coli*- and *A. baumannii*-expressing VIM-2 or SPM-1 with respect

to control cells transformed with the empty vector (see Fig. 2a). This hypervesiculation phenotype was not observed in *P. aeruginosa* cells expressing VIM-2 or SPM-1 (Fig. 2a). In the case of NDM-1, similar levels of vesicles were produced in all tested organisms, indicating that expression of NDM-1 does not affect OMV production in these bacteria. This correlates with the finding of NDM-1 in vesicles from all three organisms (Fig. 1a).

A hypervesiculation phenotype is frequently associated with an envelope cell stress response, where OMV production acts as a mechanism to alleviate conditions that compromise bacterial fitness[23,27]. We reasoned that expression of VIM-2 and SPM-1 may generate an envelope stress in *E. coli* and *A. baumannii*. We tested this hypothesis by evaluating the impact of MBLs expression on bacterial growth, as a measurement of bacterial fitness. The growth curves of *E. coli*, *P. aeruginosa* and *A. baumannii* expressing the different MBLs in the absence of selective pressure show strikingly different profiles (Fig. 2b and Supplementary Fig. 6). The expression of *bla*$_{NDM-1}$ does not entail a fitness cost in any of the three strains assayed, even at high levels of IPTG added to induce protein expression (Fig. 2b and Supplementary Fig. 6). Instead, expression of VIM-2 or SPM-1 causes a dramatic effect in the fitness of *E. coli* and *A. baumannii*, severely compromising the bacterial growth rates (Fig. 2b). In the case of *A. baumannii*, expression of VIM-2 provokes a stronger growth defect compared to SPM-1, which is evidenced by a prolonged lag period. This compromised cell metabolism would explain the lower synthesis of VIM-2 observed at this stage of growth, as mentioned above (Fig. 1a, top panel).

Larger concentrations of IPTG induce higher expression levels of VIM-2 and SPM-1 that further retard the growth of *E. coli* and *A. baumannii* (Supplementary Fig. 6). In the latter case, the

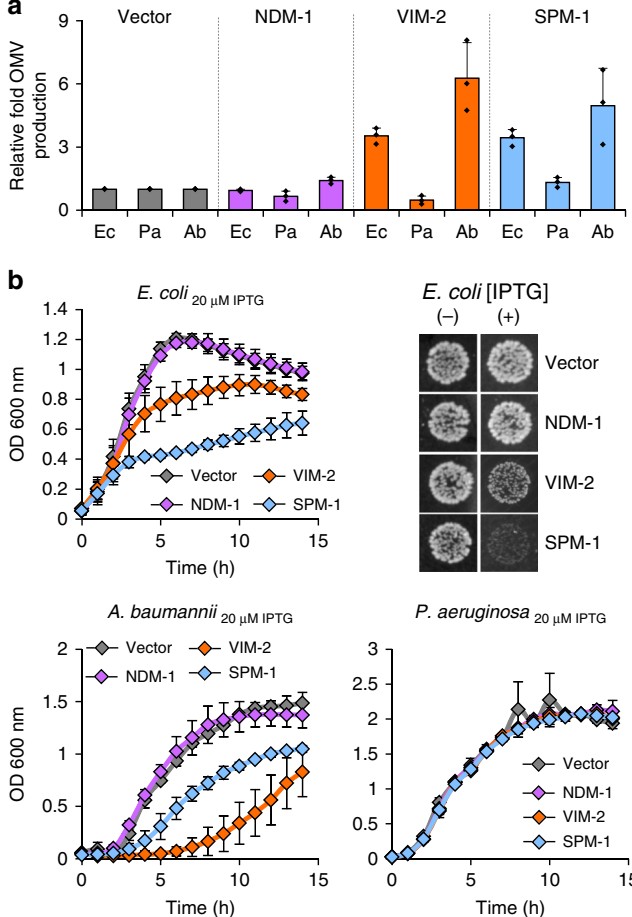

**Fig. 2** Expression of VIM-2 and SPM-1 in non-frequent hosts induces hypervesiculation and compromises fitness. **a** Relative fold OMV production in cultures of the indicated hosts (Ec, Pa, or Ab) expressing $bla_{NDM-1}$, $bla_{VIM-2}$, or $bla_{SPM-1}$ at 20 μM of IPTG. OMVs were quantitated based on total protein concentrations, normalizing to $OD_{600}$, and dividing by $OD_{600}$-normalized OMV production in control samples (corresponding to cells harboring the empty vector). Data correspond to three independent experiments and are shown as the mean value. Error bars represent standard deviations (s.d.). **b** Growth curves for *E. coli*, *A. baumannii*, and *P. aeruginosa* carrying the empty vector (Vector) or expressing $bla_{NDM-1}$, $bla_{VIM-2}$, or $bla_{SPM-1}$ in aerobic conditions in LB medium at 37 °C with 20 μM of IPTG. $OD_{600}$ of the cultures was recorded every hour for 15 h. Data correspond to three independent experiments and are shown as the mean value. Error bars represent standard deviations (s.d). *Right top panel*: bacterial viability in a spot plating assay. Overnight cultures from *E. coli* expressing the different MBLs were serially diluted and 10 μl aliquots spotted onto LB agar without (-) or with the addition of 50 μM of IPTG. Colonies were allowed to develop for 24 h at 37 °C. Source data are provided as a Source Data file

expression of VIM-2 or SPM-1 was lethal at 50 μM IPTG (Supplementary Fig. 6). The expression of SPM-1 in *E. coli* strains at 50 μM of IPTG also resulted in cell death (Fig. 2b, right top panel). In stark contrast, *P. aeruginosa* strains expressing VIM-2 or SPM-1 showed normal growth curves (Fig. 2b and Supplementary Fig. 6). This correlates with the observation that *P. aeruginosa* cells do not elicit secretion of VIM-2 or SPM-1 into OMVs (Fig. 1a). These results unveil a positive correlation between the accumulation of precursor forms of VIM-2 and SPM-1, the biological cost upon expression of these enzymes and their secretion into vesicles in *E. coli* and *A. baumannii*.

We conclude that VIM-2 and SPM-1 are not readily secreted in OMVs from their most frequent host (*P. aeruginosa*), since their expression does not compromise bacterial fitness. Instead, we observed that the expression of NDM-1 does not affect the growth of any of three bacteria tested and, moreover, is packaged in OMVs produced by all studied organisms. The differential behavior is more evident in *P. aeruginosa* strains, since their OMVs contain NDM-1 and not VIM-2 or SPM-1, despite the two latter MBLs are isolated with high frequency in this bacterium. Altogether, our results indicate that the preferential inclusion of NDM-1 in the vesicles of different microbial hosts depends on mechanisms of selective sorting associated with unique features of NDM-1, instead of being specific host issues. Moreover, we can discard the notion that NDM are eliminated into vesicles to relieve the cell from periplasmic stress. Instead, the selective secretion of VIM-2 and SPM-1 into vesicles in non-frequent hosts, associated with a hypervesiculation phenotype, discloses the presence of envelope stress in these cases.

**Expression of MBLs in non-frequent hosts induces envelope stress**. In order to understand how MBL expression induces vesicle production in non-frequent hosts, we attempted to identify the molecular mechanisms involved in this phenomenon. A hypervesiculation phenotype can be triggered by different independent mechanisms[28]. We decided to focus on the two most probable pathways, which includes: (*a*) decrease in the covalent OM-peptidoglycan cross-linking that favors OM bulging, or (*b*) the accumulation of toxic species in the bacterial envelope that are eliminated in the OMV[27]. We selected *E. coli* as a model system to study this phenomenon.

The first mechanism can be interrogated by measuring the levels of Lpp, the most abundant lipoprotein responsible for tethering the outer membrane (OM) to the peptidoglycan sacculus[27,29,30]. We purified sacculi isolated from *E. coli* DH5α strains expressing SPM-1 or VIM-2. HPLC analysis of these sacculi revealed that levels of lipoprotein (Lpp)-linked muropeptides were unaltered in both cases compared to a control *E. coli* strain not expressing MBLs (Supplementary Table 2). These results allowed us to discard this hypothesis.

The accumulation of toxic proteins in the periplasm can be evidenced by an increase in the levels and/or activity of DegP, the central housekeeping protein in the bacterial envelope that eliminates toxic species[22,31–33]. In fact, DegP and OMV production act in concert to alleviate this type of periplasmic stress[27,34]. We observed that DegP levels were increased 1.4- and 1.7-fold more in *E. coli* cells expressing VIM-2 or SPM-1, respectively, than *E. coli* cells harboring the empty vector. This finding is accompanied with a notable accumulation of *short-*DegP[35], an autoproteolysis product of DegP, revealing that DegP is activated to relieve envelope stress[35] (Fig. 3a). Instead, *E. coli* cells expressing NDM-1 do not show significant changes in DegP levels as evidence of degradation products could not be found (Fig. 3a).

The impact of the activity of DegP activity upon expression of VIM-2 or SPM-1 was tested by comparing the growth curves of the wild-type *E. coli* strain and the Δ*degP* mutant (Fig. 3b), both expressing different MBLs. The Δ*degP* strain was able to grow at similar rates than the wild-type strain in the presence of an empty vector or when expressing the non-toxic NDM-1. Instead, expression of either VIM-2 or SPM-1 was lethal to knockout *degP* cells. Taken together, these results indicate that *E. coli* experiences a severe periplasmic stress upon expression of the non-host-adapted proteins, VIM-2 or SPM-1. This stress can be relieved by means of expression of the protease DegP and by

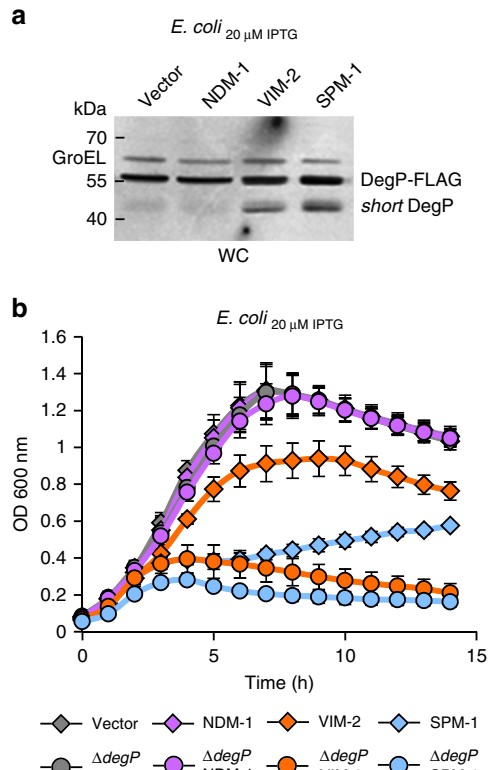

**Fig. 3** Expression of VIM-2 and SPM-1 in non-frequent hosts induces envelope stress. **a** DegP-3xFLAG protein levels by western blot, using anti-FLAG antibodies, in whole cells from *degP*-3xFLAG *E. coli* cells carrying the empty vector or expressing each MBL, at 20 μM of IPTG. *Short*- DegP is showed in cells expressing $bla_{VIM-2}$ or $bla_{SPM-1}$. This form of DegP indicates that mature DegP was self-cleaved in vivo. GroEL was used as a load control. This panel is representative of two independent experiments. **b** Growth curves of the *degP* null (Δ*degP*) and wild-type *E. coli* strains, carrying the empty vector (Vector) or expressing $bla_{NDM-1}$, $bla_{VIM-2}$, or $bla_{SPM-1}$ in aerobic conditions in LB medium at 37 °C with 20 μM of IPTG. The growth was monitored as above. Data correspond to three independent experiments and are shown as the mean value. Error bars represent standard deviations (s.d). Source data are provided as a Source Data file

vesicle-mediated elimination of the toxic MBLs that is reflected in the hypervesiculation phenotype.

**The signal peptide of MBLs determines bacterial fitness.** We sought to identify the molecular features by which expression of $bla_{VIM-2}$ or $bla_{SPM-1}$ compromise bacterial fitness in *E. coli*. In principle, this could be due to: (1) the enzymatic activity of the MBLs, (2) their cellular localization, or (3) the distinctive molecular features of each MBL. Firstly, we examined whether the carbapenemase activity was responsible of conferring the observed growth defects. Addition of a specific MBL inhibitor (bisthiazolidine compound: L-CS319[36]) to *E. coli* cells expressing VIM-2 or SPM-1 did not alleviate the MBL-mediated toxicity (Supplementary Fig. 7), allowing us to conclude that the activity of MBLs is not responsible for the microbial fitness upon expression.

Since NDM-1 is membrane-bound and VIM-2 is a soluble β lactamase, we explored the possible role of the cellular localization of MBLs on the toxicity to the host cell. Membrane anchoring in NDM-1 is coded in a lipidation sequence (lipobox) in the signal peptide. We previously determined that the cellular

localization of NDM-1 and VIM-2 can be altered by swapping the signal peptides of these proteins[17]. V2-NDM-1 is a chimera of mature NDM-1 and the N-terminus of the VIM-2, that is a soluble periplasmic protein; and N1-VIM-2 is a chimeric protein in which the sequence of mature VIM-2 is preceded by the N-terminal signal peptide of NDM-1, resulting in a membrane-anchored form of VIM-2 (Fig. 4a). The soluble variant V2-NDM-1 induced severe growth defects in *E. coli* cells, in contrast to membrane-bound NDM-1 and resembling the behavior of VIM-2 (Fig. 4b, top panel). On the other hand, membrane-anchored VIM-2, N1-VIM-2 did not elicit any growth defect (Fig. 4b, medium panel). These results could in principle suggest that membrane anchoring would prevent MBLs from being toxic in *E. coli*. MBLs with a lipobox can also be expressed as soluble proteins by mutating the Cys residue in the lipidation site (mutant C26A)[17]. Surprisingly, none of the soluble variants $NDM-1_{C26A}$ and $N1_{C26A}$-VIM-2 compromised the bacterial growth, disproving the previous hypothesis (Fig. 4b, top and medium panel). These findings suggest that the identity of the signal peptide governs the impact of MBL expression on bacterial fitness. Indeed, immunoblot analysis of *E. coli* cells expressing these variants shows clearly that those chimeras with the VIM-2 signal peptide present accumulation of the precursor form (Fig. 4b, bottom panel). In conclusion, while the cellular localization has an impact on the bacterial fitness of MBLs expression, the N-terminus of VIM-2 seems to be the main determinant of the impact of MBL expression on bacterial fitness and toxicity. Phylogenetic analysis of signal peptide sequences did not cluster according to MBLs' host range, indicating non-obvious sequence determinants defining the distribution of these enzymes among bacterial species (Supplementary Fig. 8).

We next decided to test the role of the information contained in the signal peptide sequence on bacterial fitness, by studying different soluble MBLs. We selected the soluble enzymes VIM-2, VIM-1, and IMP-1. The two natural VIM variants, VIM-2 and VIM-1 differ only by 17 residues[37–39], seven of them being located in the signal peptide. IMP-1, instead, belongs to a different family of clinically relevant, soluble MBLs, divergent in sequence respect to the VIM family[39,40]. All three proteins are expressed to similar levels in *E. coli* cells. However, cells expressing VIM-1 show a minor reduction in the growth curves (Fig. 4c, d and Supplementary Fig. 9) despite the close homology with VIM-2. Expression of IMP-1 gives rise to growth curves similar to cells carrying the empty vector, a fact that discloses the absence of fitness cost upon expression of this MBL (Fig. 4c, d and Supplementary Fig. 9).

To validate our hypothesis, we constructed a new set of chimeric enzymes by swapping the signal sequences of VIM-2 and VIM-1, and VIM-2 and IMP-1, and we measured the growth curves of *E. coli* cells expressing these variants. Substitution of the signal peptide of VIM-2 by those of VIM-1 or IMP-1 (variants V1-VIM-2 and I1-VIM-2) alleviated the fitness cost caused by expression of wild-type VIM-2 (see Fig. 4c, d). Conversely, expression of IMP-1 using the signal sequence of VIM-2 (V2-IMP-1) yielded few and very small colonies, indicating that production of this variant is defective. Indeed, expression of this variant elicited strong growth defects even when cells were grown without adding IPTG (Fig. 4c, d). The only exception is expression of VIM-1 with the peptide leader of VIM-2, which does not compromise bacterial fitness (Fig. 4c, d). Finally, all cases of cells with growth defects expressing these variants showed accumulation of the precursor forms (Fig. 4c, bottom panel). We conclude that the signal peptide of VIM-2 is responsible of eliciting envelope stress in *E. coli* by accumulation of the precursor protein that is toxic to the cell and compromises bacterial fitness.

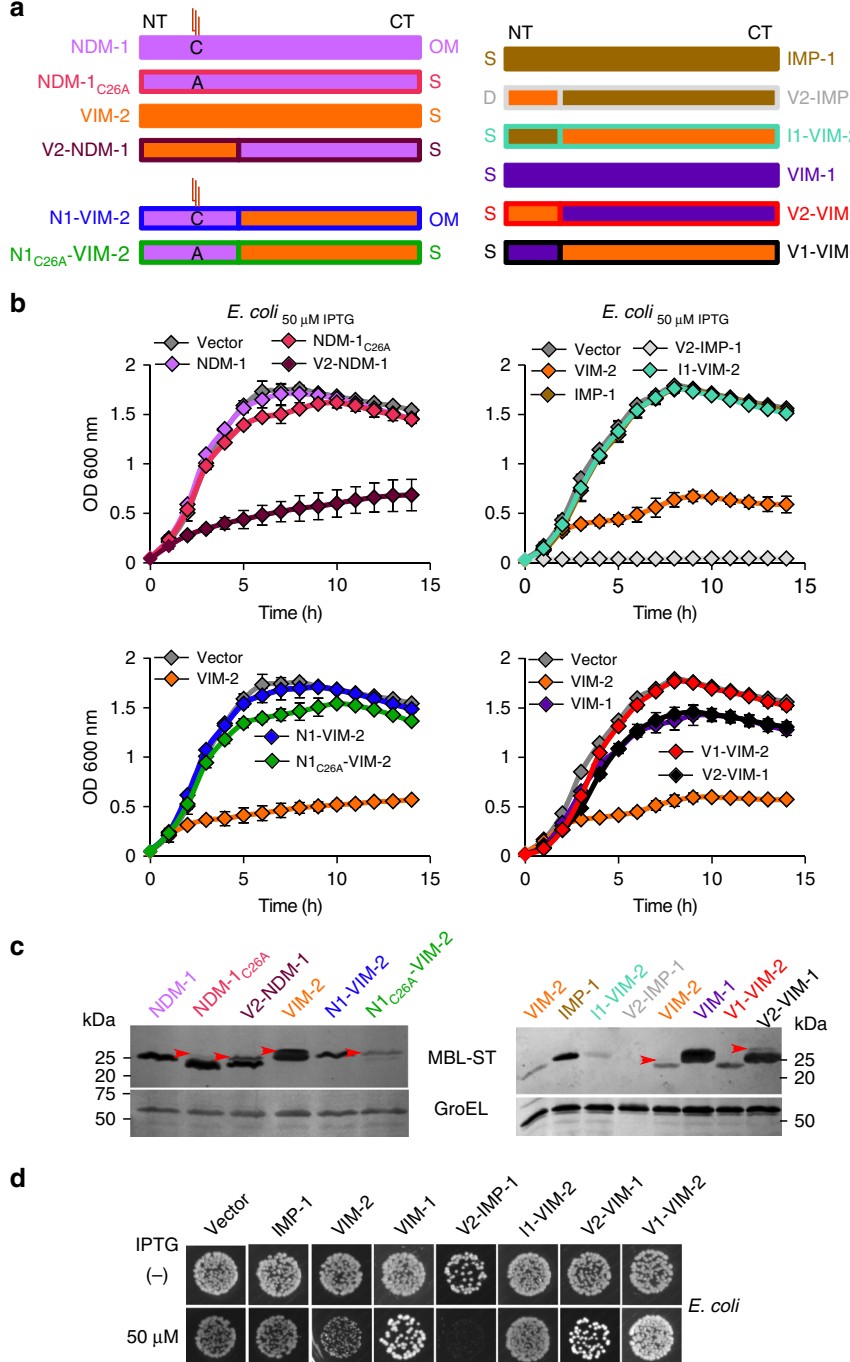

**Fig. 4** Impact of cellular localization and protein features of the MBLs on fitness cost. **a** Schematic representation of the chimeric MBLs variants with altered cellular localizations and engineered signal peptides. *NT*: N-terminus and *CT*: C-terminus. *OM*: protein lipidated and anchored to outer membrane. *S*: soluble periplasmic protein. *C*: cysteine anchoring residue into lipobox sequence. *A*: alanine residue. By mutation of the cysteine residue by an alanine residue, C26A, the lipidation site is removed resulting in a soluble variant. The same color coding is used for each variant throughout this Figure. **b** Growth curves of *E.coli* strains expressing *bla*$_{NDM-1}$ or *bla*$_{VIM-2}$ with altered cellular localizations in LB at 37 °C with 50 μM of IPTG (*top and medium panels*). The growth was monitored as above. Data represent the mean value (±s.d) of three independent experiments. *Bottom panel*: Protein levels of NDM-1, VIM-2 and their chimeras with altered cellular localizations by immunoblotting in whole cells from *E. coli* expressing each MBL after induction at 20 μM with IPTG. Red arrows indicate the precursor forms. **c** Growth curves of *E. coli* strains expressing *bla*$_{IMP-1}$, *bla*$_{VIM-2}$, and their chimeras with their exchanged signal peptidesI1-VIM-2 and V2-IMP-1 (*top panel*). Growth curves of *E. coli* strains expressing *bla*$_{VIM-1}$, *bla*$_{VIM-2}$, and their chimeras with their exchanged signal peptides (*medium panel*). *Bottom panel* shows the protein levels of VIM-2, VIM-1, and their chimeras with their signal peptides swapped, by immunoblot analysis. Lower panels show loading control (GroEL). **d** Bacterial viability in a spot plating assay. Overnight cultures from *E. coli* expressing VIM-2, IMP-1, VIM-1, and their chimeras were serially diluted and 10 μl aliquots spotted onto LB agar without (−) or with the addition of 50 μM of IPTG (+). Colonies were allowed to develop for 24 h at 37 °C. Panels **c** (*bottom panel*) and **d** are representative of two independent experiments. Source data are provided as a Source Data file

## Discussion

Gram-negative pathogens expressing MBLs seriously threaten public health. Plasmid-encoded MBLs are disseminating worldwide at an alarming pace among these pathogens, conferring resistance to carbapenems and limiting the usefulness of these last-resort antibiotics. Many elegant studies have shown a strong relationship between the rapid spread of MBL genes and the presence of conjugative plasmids[41–44]. However, the rapid spread of the $bla_{NDM-1}$ gene in a short period of time in multiple bacterial hosts could not solely be attributed to the genetic platform harboring this gene. Despite *K. pneumoniae* and *E. coli* are the most frequently described NDM-1-producing bacteria[45,46], the $bla_{NDM-1}$ gene has also been found in non-fermentative bacteria including *P. aeruginosa* and *Acinetobacter* spp.[47], and in a number of environmental organisms, such as *Enterobacter asburiae*, *Providencia rettgeri*, and *Acinetobacter baylyi*. In contrast, the host preferences of VIM alleles are very diverse. VIM-2, the most prevalent MBL, despite being present in plasmids of broad host range[48], has been mostly detected in *P. aeruginosa* isolates[9,49]. There are exceptional reports of Enterobacteriaceae members, such as *E. coli* and *Serratia marcescens*, producing VIM-2[50,51]. These particular strains, however, may have co-evolved to endure the toxicity associated with VIM-2 expression[52]. The variant VIM-1 (differing by only 17 residues with VIM-2), instead, is more frequently isolated in Enterobacteriaceae (such as *E. coli* ST 131) and non-fermenters[9,53,54]. Finally, SPM-1 is exclusively confined to *P. aeruginosa*[9]. Surprisingly, the resistance phenotype of NDM-1, VIM-2 and SPM-1 in different hosts does not correlate with these host preferences, and little is known about the mechanisms that determine this puzzling host specificity.

In the present study, we report how the molecular features of MBL proteins elicit different fitness costs when expressed in bacteria under permissive conditions (i.e., in the absence of the evolutionary pressure of antibiotics). We identify this fitness cost as a selection driver that determines the dissemination of MBLs and their host specificity. This fundamental property does not only apply to differentiate distinct MBL families, since subtle changes such as few mutations among allelic variants elicit drastic effects in fitness cost upon MBL expression, impacting on host specificity. Based on this paradigm, we show that NDM-1 is uniquely tailored to be expressed without inducing any fitness cost in different bacterial hosts, in contrast to other MBLs.

The soluble lactamases VIM-2 and SPM-1, instead, induce a high fitness cost by compromising bacterial growth when expressed in *A. baumannii* and *E. coli*, which are not frequent hosts for these MBLs. VIM-2 and SPM-1 are capable of conferring resistance to β-lactam in non-frequent hosts, being able to outperform their resistance profile in *P. aeruginosa* in some cases (Fig. S1 and Table S1). Thus, host specificity cannot be explained only based in the resistance phenotype of MBLs in different hosts. Previous studies have shown that the fitness cost under permissive conditions compromises the persistence of an antibiotic resistance determinant in a specific bacterial host[55–57]. In the particular case of the carbapenemase SME-1, Marciano et al.[55] have shown that mutations in its signal peptide alter the fitness cost of protein expression. Our results show that this effect has been dominant in the distribution and host specificity of MBLs.

Reduction of the bacterial growth is accompanied in all cases by accumulation of the precursor protein in non-frequent or less common hosts. Instead, expression of either VIM-2 or SPM-1 in their frequent host *P. aeruginosa*, does not imply any burden, an observation that is associated with the absence of accumulation of the precursor protein. We demonstrate that accumulation of precursor MBLs entails an envelope stress. This conclusion is supported by the finding of growth defects, a hypervesiculation

phenotype and a periplasmic stress response, as evidenced by the activation of the housekeeping protease DegP. We hypothesized that accumulation of the toxic precursor forms could be due to the sequence of the signal peptides, which may lead to deficient translocation and processing events in some hosts. We demonstrate that this is the case by engineering chimeric proteins that allowed the expression of several MBLs with other signal peptides. Expression of VIM-2 with the signal peptide of other soluble MBLs (such as VIM-1 or IMP-1) minimizes or even abolishes the growth defects observed when expressing this MBL in *E. coli* and *A. baumannii*, revealing that these leader peptides are more efficiently processed by the protein export machinery in these hosts. Expression of VIM-1 in *E. coli* and *A. baumannii* impaired bacterial growth to a minor extent, unlike expression of VIM-2, which can be lethal. These results demonstrate that the signal peptide sequence is a main determinant of the toxicity of a given MBL in different bacterial hosts.

Most VIM alleles can be classified into two groups deriving from VIM-1 and VIM-2. The host prevalence reported for the different $bla_{VIM}$ genes resembles those of the parent gene, confirming this hypothesis. VIM-12 is a particular case, being a VIM-1/VIM-2 hybrid that results from the N-terminal sequence from VIM-1 (including the signal peptide) and the C-terminus from VIM-2[58]. Despite being present in genetic cassettes similar to those of VIM-2, the $bla_{VIM-12}$ gene has been reported only in Enterobacteriaceae[58], revealing that the signal peptide is more important than the genetic environment of the MBL gene in determining the host specificity. Moreover, this sequence can be used to predict the host specificity and possible spread of new MBL alleles.

Type I signal peptidases (SPase I) are responsible of cleaving the signal peptides from the secretory precursor proteins during translocation across the cytoplasmic membrane[59–61]. Most Gram-negative bacteria possess a single SPase I[62], while *Pseudomonas aeruginosa* is an exception in expressing *two* signal peptidases[63]. It would be interesting to assess in future work whether incorporation of an additional SPase I, or an increase in endogenous SPase I levels may help to alleviate the stress caused by accumulation of toxic precursors. In any case, the saturation of the protein export machinery to the cell envelope is responsible for the fitness cost triggered upon expression of these enzymes.

Processing of NDM-like enzymes follows a different pathway. NDM β-lactamases are lipoproteins anchored to the outer membrane, and their posttranslational processing requires the action of signal peptidase II (SPase II) that recognizes the lipobox sequence and cleaves the signal peptide[16,64]. Since membrane anchoring provides several evolutionary advantages to NDM-1, we tested whether its cellular localization could be responsible of the lack of fitness cost imposed by expression of this enzyme. However, expression of the soluble variant NDM-1 C26A was not toxic to *E.coli* cells, allowing us to discard the notion that membrane anchoring could be responsible of the adaptability of NDM to different hosts. Instead, expression of NDM-1 with the leader peptide of VIM-2 lead to precursor accumulation and associated growth defects. Overall, these results highlight the crucial role of the signal peptide in defining the adaptability of different MBLs to particular bacterial hosts.

Production of OMVs is a mechanism that helps alleviating envelope stress by eliminating toxic proteins from the periplasm of Gram-negative bacteria. This is the case for VIM-2 and SPM-1 in non-frequent hosts, such as *E. coli* and *A. baumannii*. Our assertion is supported by the finding that these proteins are not packaged into OMVs from its most frequent host, *P. aeruginosa*. NDM-containing vesicles show higher specific activity than OMVs carrying VIM-2 or SPM-1 (Supplementary Fig. 4c). Since MBLs are active when they are properly folded and loaded with

Zn(II), we conclude that a fraction of VIM-2 and SPM-1 secreted into vesicles is not properly folded. Indeed, the lower specific activity of SPM-1 into *E. coli* and *A. baumannii* vesicles indicate a larger accumulation of misfolded protein, in line with the more restricted host specificity of this MBL compared to VIM-2. These results are in line with the observation that the accumulation of toxic misfolded porins in the periplasm of *E. coli* induces envelope stress that is partially relieved by secretion of these misfolded proteins in OMVs[34].

NDM-1 is exported to vesicles in *E. coli*, *P. aeruginosa* and *A. baumannii*, a process that cannot be correlated to any envelope stress caused by protein expression. Indeed, expression of NDM-1 as a membrane-bound protein did not compromise bacterial growth in any of the bacteria tested here, and exhibited adequate posttranslational processing. We conclude that expression of NDM-1 is favored in many microorganisms due to the lack of fitness cost, accounting for the broad host range of this MBL. These results also reveal that NDM-1 is tailored to be selectively secreted into OMVs. NDM-1-containing vesicles showed a high carbapenemase activity, i.e., NDM-1 is secreted as a folded and active protein. This selective secretion of NDM-1 β-lactamase into vesicles may then serve another purpose.

Secretion of NDM-1 into vesicles has been proposed as a mechanism that may help disseminating the *bla*$_{NDM}$ gene[17]. This secretion implies an advantage to the organism under the presence of antibiotics by disseminating the carbapenemase activity beyond the bacterial cell[17], thus titrating the available antibiotic at the infection site. At the same time, these vesicles are able to protect populations of otherwise susceptible bacteria, favoring the opportunity of uptaking the *bla*$_{NDM}$ gene by horizontal gene transfer or by the OMVs themselves[21]. This could be crucial in polymicrobial infections, resulting in lethal treatment failure. The current results strongly support the role of OMVs in the dissemination of NDM-1. This enzyme clearly exhibits many evolutionary advantages attributable to its protein features compared to other clinically relevant MBLs, such as VIM-2 and SPM-1. Since the signal peptide is highly conserved in all NDM alleles, we anticipate that these features are preserved in all proteins from the NDM family.

Bacteria use a toolkit for acquiring, maintaining and disseminating resistance that involves different features and has been shaped by evolution. On one hand, the intrinsic features of the genetic mobile elements coding for resistance genes impact on the rate of dissemination of the *bla* genes. On the other hand, the β-lactamase activity of the different proteins acts as a selection driver of certain alleles that are better adapted to some hosts under the evolutionary pressure of specific β-lactams. This is the case of SPM-1, whose active site is finely tailored by unique mutations to hydrolyze anti-pseudomonal antibiotics, thus being an enzyme restricted to *P. aeruginosa* strains[10]. Also, the ceftazidimase activity of VIM proteins has been shown to be a selection driver of this family of MBLs[65]. In addition to these well-known mechanisms, here we provide compelling evidence that the signal peptide is a key protein determinant in MBLs that defines the host adaptability in the absence of antibiotics based on the fitness cost resulting from its expression.

Antibiotic resistance affects bacterial fitness under permissive conditions. This fitness cost compromises the chances of acquiring and maintaining genes able to confer resistance under more stringent conditions. Thus, bacteria must deal with a balance between the gain of function, that is relevant for survival to face the challenge of antibiotics, and the loss of function upon the deleterious effect of MBL production in the absence of antibiotic selective pressure. We propose that the lack of fitness cost of expressing NDM-1 and its secretion into OMVs are two selection drivers that have favored the worldwide and broad

host range dissemination of this β-lactamase, boosting the impact of its resistance phenotype. This paradigm allows us to account for the epidemiology of the different MBLs. Knowledge of the biological cost of expressing MBLs is of relevance not only to anticipate resistance dissemination among different hosts, but also to evaluate clinical interventions that may help in reducing resistance.

## Methods

**Bacterial strains and growth conditions.** The representative species for this study were the following strains: *Acinetobacter baumannii* ATCC 17978 [Ab], *Pseudomonas aeruginosa* PAO1 [Pa] (both representatives of non-fermenter organisms) and *Escherichia coli* DH5α [Ec] (as representative of *Enterobacteriaceae*). For each organism studied, a set of isogenic clones expressing different wild-type MBL genes (*bla*$_{NDM-1}$, *bla*$_{VIM-2}$, *bla*$_{SPM-1}$, *bla*$_{IMP-1}$, and *bla*$_{VIM-1}$) and chimeras (*bla*$_{NDM-1C26A}$, *bla*$_{V2-NDM-1}$, *bla*$_{N1-VIM-2}$, *bla*$_{N1C26A-VIM-2}$, *bla*$_{I1-VIM-2}$, *bla*$_{V2-IMP-1}$, *bla*$_{V1-VIM-2}$, and *bla*$_{V2-VIM-1}$) was constructed. *E. coli* DH5α and *P. aeruginosa* PAO1 were used for expression of empty vector pMBLe and also for expression of the different MBLs already cloned in the pMBLe[17]. *A. baumannii* ATCC 17978 was used for expression of plasmids pMBLe-OA and pMBLe-OA-*bla*$_{MBLs}$, constructed in this work as explained below. Cells were routinely grown aerobically at 37 °C in Luria–Bertani (LB broth) or on LB agar plates, except when indicated. Gentamicin was used when necessary at 20 μg ml$^{-1}$. All reagents and chemicals were from Sigma-Aldrich, except the LB culture media that were from Difco, and oligonucleotides and enzymes that were from Life Technologies.

**Plasmid and strain construction.** Plasmid isolation, DNA purification, restriction enzyme digestion, ligation, and transformation were performed by standard methods according to Sambrook et al.[66]. For expressing the different *bla* genes either in *E. coli* or *P. aeruginosa*, constructions of the MBL variants already cloned in the pMBLe[17] were used, which contain the full-length *bla*$_{MBLs}$ genes, including their native signal peptides, fused to a C-terminal Strep-tag II sequence under the control of an β-IPTG-inducible pTac promoter. To clone and express *bla*$_{VIM-1}$, the same strategy was used as for the other MBLs[17]. Briefly, the full-length *bla*$_{VIM-1}$ gene was PCR-amplified from a genomic preparation of clinical isolate producing VIM-1 (*E. coli* M13488), kindly provided by Dr. Roberto Melano (Public Health Ontario Lab, Canada), using primers VIM1NdeI$_{Fw}$ (5′-GACATCATATGTTAAA AGTTATTAGTAGTTTATTGGTC-3′) and VIM1StHindIII$_{Rv}$ (5′-GACGTAAGC TTCTACTTTTCGAATTGTGGGTGAGACCACTCGGCGACTGAGCGATTTT TGTG-3′)

The pMBLe plasmid can efficiently replicate in *E. coli* and *P. aeruginosa*[17], but did not in *A. baumannii*. For this, the pMBLe was modified by cloning 1337 bp *ori* region from pWH1266 into AgeI restriction site of pMBLe using primers oriAbAgeI$_{Fw}$ (5′- ATATACCGGTGATCGTAGAAATATCTA-3′) and oriAbAgeI$_{Rv}$ (5′-ATATACCGGTGGATTTTAACATTTTGC-3′), obtaining the pMBLe-OA plasmid, replicative in *A. baumannii*. The pWH1266, an *Acinetobacter* genus specific plasmid[67], and *A. baumannii* ATCC 17978 strains were kindly provided by Dra. Alejandra Mussi (CEFOBI-CONICET, Argentina). Through subcloning by restriction enzyme digest, using *Bam*HI and *Hind*III, except in the case of *bla*$_{IMP-1}$, in which it was used *Bam*HI and *Eco*RI, we subcloned each *bla*$_{MBL}$ gene in the pMBLe-OA.MBL expression was induced with low concentrations of IPTG (10–50 μM), as indicated.

The construction of the *degP*-3xFLAG Km$^R$ strain was performed as described[68]. Briefly, *E. coli* DH5α containing plasmid pKD46[69] was transformed with a PCR product generated using primers degP-3xFLAG$_{Fw}$ (5′-CATTCAGCGCGGCGAC AGCACCATCTACCTGTTAATGCAGGACTACAAAGACCATGACGG-3′) and degP-3xFLAG$_{Rv}$ (5′-GGAGAACCCCTTCCCGTTTTCAGGAAGGGGTTGAGGG AGACATATGAATATCCTCCTTA-3′) and plasmid pSUB11 as template[68]. Once obtained, *E. coli* cells carrying the chromosomal *degP*-3xFLAG allele were transformed with the empty vector or the vector harboring each *bla*$_{MBL}$. To determine the intracellular levels of the DegP protein in cells expressing the different MBLs, western blot analyses were performed as described previously[17], with mouse anti-FLAG monoclonal antibodies (clone M2, Sigma-Aldrich) and immunoglobulin G-alkaline phosphatase conjugates (at a 1:3000 dilution). Protein band intensities were quantified from polyvinylidene difluoride (PVDF) membranes with ImageJ software[70]. Uncropped images of the western blot are provided in the Source Data file. The PageRuler Plus Prestained Protein Ladder (Thermo Scientific #26616) provided molecular weight standards for Figs. 1a and 3a. The Precision Plus Protein Standards Kaleidoscope (BioRad #161–0375) provided molecular weight standards for Figs. 1b and 4c, as shown in the Source Data file.

The strain Δ*degP* carrying gene deletion in *degP*, derivative of *E. coli* DH5α, was generated by Lambda Red-mediated recombination using previously protocols[69] and the primers degP-P1 (5′-ACAGCAATTTTGCGTTATCTGTTAATCGAGA CTGAAATACgtgtaggctggagctgcttcg-3′) and degP-P2 (5′-GGAGAACCCCTTCCC GTTTTCAGGAAGGGGTTGAGGGAGAcatatgaatatcctcctta-3′).

**Construction of MBL mutants.** The different MBLs variants with their signal peptides swapped (I1-VIM-2, V2-IMP-1, VIM-V1-VIM-2, and V2-VIM-1)

were constructed by overlap extension PCR using overlapping primers VIM-2$_{Fw}$ (5′-AGTCCGCTCGCTTTTTCCGTAGATTC-3′), VIM-1$_{Fw}$ (5′-AGTCCGTTAG CCCATTCCGGGGAGCC-3′), IMP-1$_{Fw}$ (5′-TCTTTGCCAGATTTAAAAATTG AAAAGC-3′), V2-IMP-1$_{Rv}$ (5′-GCTAGAATCTACGGAAAAAGCGAGCGG ACTCTCTGCTGCGGTAGCAATG-3′), V2-VIM-1$_{Rv}$ (5′-GCTAGAATCTAC GGAAAAAGCGAGCGGACTTTGCGACAGCCATGACAG-3′), I1-VIM-2$_{Rv}$ (5′-AAGCTTTTTCAATTTTTAAATCTGGCAAAGACGCAATAGCCATGATAG-3′) and V1-VIM-2$_{Rv}$ (5′-ACTCGGCTCCCCGGAATGGGCTAACGGACTCGCAA TAGCCATGATAG-3′) with external primers pMBLe$_{Fw}$ (GCTGTTGACAATTA ATCATCGGTC) and pMBLe$_{Rv}$ (CACTACCATCGGCGCTACG); employing the same strategy previously used by us for the generation of the NDM-1 or VIM-2 variants with altered cellular localizations[17]. The PCR-amplified genes were sub-cloned into plasmid pMBLe in E. coli strains. All constructs were verified by DNA sequencing (University of Maine, USA).

**MBL detection and β-lactamase activity measurements.** MBL protein levels were determined by SDS-PAGE followed by western blotting with Strep-tag II monoclonal antibodies (at a 1:1000 dilution from 200 µg ml$^{-1}$ solution) (Novagen) and immunoglobulin G-alkaline phosphatase conjugates (at a 1:3000 dilution). Briefly, the samples were mixed with loading buffer and heated to denature the peptide structure. SDS-PAGE (14%) was used for separation of the sample components and subsequently transferred onto a PVDF membrane (GE). Western blotting with antibodies detecting GroEL was performed as a loading control. β-lactamase activity was measured in a JASCO V-670 spectrophotometer at 30 °C in 10 mM HEPES, 200 mM NaCl pH 7.4 (for OMVs), in 0.1 cm cuvettes, using 400 µM imipenem as a substrate. Imipenem hydrolysis was monitored at 300 nm ($\Delta\varepsilon_{300nm}$ = −9000 M$^{-1}$ cm$^{-1}$). OMVs samples were normalized according to total protein concentrations, as quantified with the Pierce® BCA Protein Assay Kit (Thermo Scientific). Uncropped images of the western blot are provided in the Source Data file.

**Cell fractionation and proteolysis experiments.** E. coli cells were grown at 37 °C until OD of 0.5, induced with 20 µM of IPTG for expression of VIM-2 or SPM-1 and growth continued for two hours. Cells were fractioned into periplasm and spheroplasts by the EDTA-Lyzosime method as previously reported[17]. Briefly, E. coli cultures were pelleted and cells were washed once with 20 mM Tris, 150 mM NaCl, pH 8.0. The washed cells were resuspended in 20 mM Tris, 0.1 mM EDTA, 20% w per v sucrose, 1 mg ml$^{-1}$ lysozyme (from chicken egg white, Sigma-Aldrich, protein ≥ 90%), 0.5 mM PMSF, pH 8 (resuspension volume was normalized according to the formula $V = 100 \mu L \times OD_{600} \times V_c$, where $V_c$ is the starting volume of culture sample), incubated with gentle agitation at 4 °C for 30 min, and finally pelleted, with the periplasmic extract in the supernatant. A fraction of spheroplasts were resuspended in 20 mM Tris, 150 NaCl, pH 8.0, lysed by sonication and separated into soluble and insoluble fractions by centrifugation at 12,000 × g for 20 min at 4 °C. The remaining fraction of spheroplasts was resuspended in 20 mM Tris, 10 mM CaCl$_2$, pH 8.0 and treated with 50 µg ml$^{-1}$ Proteinase K for 30 min at 0 °C.

**MIC determinations.** Imipenem (IMI), ceftazidime (CAZ), cefepime (CFP), and piperacillin (PIP) MIC determinations were performed in LB medium using the agar macrodilution method according to CLSI guidelines[71]. In all cases, $bla_{MBL}$ expression was induced with 10 µM IPTG.

**Purification of OMVs.** OMVs were purified from early stationary phase cultures of E. coli, P. aeruginosa, and A. baumannii. Briefly, 250 ml of LB medium was inoculated with 3 ml of saturated E. coli, P. aeruginosa or A. baumannii pMBLe-bla culture, grown at 37 °C up to OD$_{600}$ of 0.4, induced with 20 µM IPTG, and growth continued overnight with agitation for the case of E. coli strains. In the case of P. aeruginosa and A. baumannii strains, after IPTG addition, the growth was continued for 4 h at 37 °C[17]. The cells were harvested and the supernatant filtered through a 0.45-µm membrane (Millipore). Ammonium sulfate was added to the filtrate at a concentration of 55% w per v, followed by overnight incubation with stirring at 4 °C. Precipitated material was separated by centrifugation at 12,800 x g for 10 min, resuspended in 10 mM HEPES, 200 mM NaCl, pH 7.4, and dialyzed overnight against > 100 volumes of the same buffer. Next, the samples were filtered through a 0.45-µm membrane, layered over an equal volume of 50 % w per v sucrose solution and ultracentrifuged at 150,000 × g for 1 h and 4 °C[17]. Pellets, containing OMVs, were washed once with 10 mM HEPES, 200 mM NaCl, pH 7.4, and stored at –80 °C until use. OMVs were quantified by total protein dosage with the Pierce® BCA Protein Assay Kit (Thermo Scientific). These quantifications were comparative for each bacterium given that the levels of MBLs in vesicles were insignificant with respect to total protein amounts.

Quality of vesicle preps was assessed by negative staining transmission electron microscopy, as described elsewhere[20]. Briefly, vesicles were fixed in 4% glutaraldehyde in 10 mM HEPES, 200 mM NaCl, pH 7.4 for 30 min at room temperature, stained with 2% uranyl acetate and visualized in a Zeiss EM 109T Transmission Electron Microscope. Finally, the absence of cellular contaminants due to cell lysis during culture growth or cell manipulation was verified by

immunodetection of cytoplasmic GroEL in OMV preps. GroEL antibodies were kindly provided by Dr. Alejandro Viale (IBR-CONICET, Argentina).

**Growth curves.** Effects of MBLs expression on bacterial growth were obtained by measuring cell density at OD$_{600}$ every 60 min in a BioTek Synergy two multimode microplate reader for 15 h at 37 °C. All bacterial strains were grown in LB broth, supplemented with gentamicin, at 37 °C and 200 rpm. Overnight cultures were then diluted 1:100 in LB and applied in duplicate into a sterile 96-well microplate (Greiner Bio-one) with gentamicin at 20 µg ml$^{-1}$ and either with or without addition of different concentrations of IPTG, as indicated. For evaluating the role of carbapenemase activity on fitness cost, the growth curves were realized in the absence and presence of an enzymatic activity inhibitor. For this, a bisthiazolidine (L-CS319) was used at a concentration that ensured the complete inhibition of MBLs[36]. All the data represent an average of at least three independent experiments.

**Bacterial viability by spot plating assay.** To estimate the ability of bacterial cells to survive and form a colony according to IPTG concentration, overnight cultures were serially diluted and 10 µl aliquots spotted onto LB agar containing gentamicin at 20 µg ml$^{-1}$ and different concentrations of IPTG or no IPTG, as indicated. Colonies were allowed to develop for 24 h at 37 °C before photographical recording.

**Multiple sequence alignment and phylogenetic trees.** Signal peptide sequences were aligned with the homology extention (PSI-Coffee) multiple sequence alignment tool available at http://www.tcoffee.org/. Phylogenetic trees were constructed (LG substitution model and 100 bootstraps) using the maximum likelihood algorithm PhyML (http://www.phylogeny.fr), and unrooted trees were drawn with DrawTree from the Phylip package.

**Preparation of peptidoglycan.** Peptidoglycan was prepared, as previously described[72] from cultures of E. coli grown in LB medium at 37 °C with aeration. Cells were grown overnight in LB medium with the appropriate antibiotic and then cultured 1:100 into 300 ml of LB medium up to OD$_{600}$ of 0.4, induced with 20 µM IPTG, and growth continued for several hours until an OD$_{600}$ of 1.5 was reached. The cells were then incubated on ice until the temperature of the culture was 4 °C. Samples were centrifuged at 10,000 × g for 10 min at 4 °C, resuspended in 3 ml of distilled H$_2$O, and slowly mixed drop by drop with an equal volume of 6% (wt/vol) boiling SDS with vigorous stirring until the mixture was homogeneous. The cells were boiled for 5 min, transferred into 10-ml glass tubes, and heated at 90 °C for 24 h in a Baxter heat block. Sacculi were concentrated by centrifugation at 65,000 rpm for 15 min at 20 °C in an ultracentrifuge (Beckman-Optima TLX with a TLA110 rotor). The pellet was washed with water until no SDS was detected, according to the method of Hayashi[73]. The last pellet from the washing procedure was suspended in 1 ml of 10 mM Tris-HCl (pH 7.2) and digested first with 100 µg ml$^{-1}$ α-amylase (EC 3.2.1.1; Sigma-Aldrich, St. Louis, MO) for 1 h at 37 °C and then with 100 µg ml$^{-1}$ preactivated pronase E (EC 3.4.24.4; Merck, Darmstadt, Germany) at 60 °C for 90 min. The enzymes were inactivated by boiling for 20 min in 1% (final concentration) SDS. The cell walls were collected by centrifugation as described above and washed three times with water. Peptidoglycan was stored in water at 4 °C.

**Preparation and separation of muropeptides.** Peptidoglycan was digested in 50 mM phosphate buffer (pH 4.9) with Cellosyl (Hoechst AG, Frankfurt, Germany), at a 100 µg ml$^{-1}$ final concentration, at 37 °C overnight. The enzyme was inactivated by boiling the sample for 2 min in a water bath and centrifugation (Eppendorf centrifuge at maximum speed for 10 min) to remove insoluble debris. The supernatant was mixed with a 1/3 volume of 0.5 M sodium borate buffer (pH 9.0) and reduced with excess sodium borohydride (NaBH$_4$) for 30 min at room temperature. The pH was tested with pH indicator strips (Acilit; Merck) and adjusted to pH 3 with orthophosphoric acid. All samples were filtered (Millex-GV filters with a 0.22 µm pore size and a 2.5 mm diameter; Millipore, Cork, Ireland) and stored at −20 °C. Separation of the reduced muropeptides was performed essentially as described previously by Glauner et al.[74,75]. Muropeptides were analyzed by using a binary-pump Waters high-performance liquid chromatography (HPLC) system (Waters Corporation) fitted with a reverse-phase RP18 Aeris peptide column (250 by 4.6 mm with a 3.6-µm particle size; Phenomenex, USA) and a dual-wavelength absorbance detector (Waters UV-1570 spectrophotometer). The eluted muropeptides were monitored by measuring the absorbance at 204 nm. When required, the individual peaks were collected, vacuum dried, and stored at −20 °C.

**Quantification of muropeptides.** Individual muropeptides were quantified from their integrated areas (Waters Breeze) by using samples of known concentrations as standards. Concentrations of the standard muropeptides were determined as described previously by Work[76]. The average PG chain length was calculated by dividing the molar amount of anhydro-muropeptide (chain termini) by the total molar amount of muropeptides in muramidase-digested PG, and the degree of

cross-linking was calculated by calculating the molar ratio of dimers and trimers to total muropeptide, as described previously by Glauner et al.[74,75]. In each experiment, muropeptides were quantified from two biological replicates, and the average molar ratio (mol%) values are presented (see Supplementary Table 2). Experimental variation resulted in changes for each muropeptide measurement of ≤ 5%.

**Reporting summary**. Further information on research design is available in the Nature Research Reporting Summary linked to this article.

## Data availability
The data supporting the findings of this study are available within the article and its Supplementary Information files. Source data for figures are provided as a Source Data file.

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

## Acknowledgements

We thank Dra. M. Alejandra Mussi (CEFOBI, CONICET-UNR, Argentina) for providing plasmid pWH1266 and *A. baumannii* ATCC 17978 strain, Dr. Roberto Melano (Public Health Ontario Lab, Canada) and Dr. Fernando Pasterán (INEI-ANLIS, Argentina) for providing clinical strain producing VIM-1, and Alejandro Viale (IBR, CONICET-UNR, Argentina) for GroEL antibodies. We are grateful to Marina Avecilla for excellent technical assistance and we thank Agustina Rossi for technical guidance regarding bis-thiazolidine assays. This research was supported by grants from the National Institutes of Health (R01AI100560 to R.A.B. and A.J.V., R01AI063517 to R.A.B., and R01AI072219 to R.A.B.), Agencia Nacional de Promoción Científica y Tecnológica (ANPCyT) to A.J.V. This study was supported in part by funds and/or facilities provided by the Cleveland Department of Veterans Affairs, Award Number 1I01BX001974 to R.A.B. from the Biomedical Laboratory Research & Development Service of the VA Office of Research and Development and the Geriatric Research Education and Clinical Center VISN 10 to R.A.B. L.J.G. and A.J.V. are staff members of CONICET and C.L. is recipient of a postdoctoral fellowship from ANPCyT. Research from JAA lab was supported by grants BFU2009-09200 and IPT2011-0964-900000 from the Spanish Ministerio de Ciencia e Innovación, and DIVINOCELL FP7 HEALTH-F3-2009-223431 from the European Commission.

## Author contributions

C.L. performed the microbiological, molecular biology, and biochemical experiments. J.A.A. performed the peptidoglycan analysis and muropeptides quantification. C.L., J.A.A., L.J.G., R.A.B. and A.J.V. analyzed and discussed data. C.L., L.J.G. and A.J.V. wrote the paper. All authors discussed the results and commented on the manuscript. The content is solely the responsibility of the authors and does not necessarily represent the official views of the National Institutes of Health or the Department of Veterans Affairs.

## Additional information

**Competing interests:** The authors declare no competing interests.

