## [Peer Review File · Nature Communications]

Reviewers' comments:

Reviewer #1 (Remarks to the Author):

Protein determinants 1 of dissemination and host specificity of metallo β lactamases

This extremely interesting and inspiring piece of work indicates that the bacterial host range of MBLs is determined by the impact of MBL protein expression on bacterial fitness. In particular, the authors show the critical role of the protein signal peptide sequence in the production/alleviation of the envelope stress. Membrane vesiculation reduces such stress, encapsulating non-host-adapted MBLs. Many of these findings are really new, and of great potential interest to understand several key-processes in the emergence and epidemiology of antibiotic resistance.

Line 59. "Clinical inhibitors" is not a precise term. I suggest: "Even though there is an active research on this field, inhibitors of these enzymes, largely distributed in Gram negative clinical strains, are not available for therapy"

Line 81: *Enterobacter cloacae*

Line 83-84. References 17 or 18 does not mention at all horizontal gene transfer. The benefit of being into membranes relates with the ability of functioning with small levels of zinc.

It is not clear why NDM-containing vesicles can promote horizontal gene transfer, as NDM genes are not in the vesicle. I can make two hypothesis: i) NDM-vesicles released from the cell with NDM-genes, can fuse with the membranes of antibiotic susceptible cells, and then helping them to resist to carbapenems. That will be cool, but need to be demonstrated. ii) the NDM vesicles can contain plasmids with NDM-genes, and thus these vesicles ALSO promotes transfer. But to my knowledge that also should be demonstrated.

Line 93. What is the meaning of "non-native bacteria"? The reader might understand that VIM-1 and SPM-1 have evolved into *P. aeruginosa* (or maybe related species). Any support for this assumption?

Lines 100-102. Is that also the case for other metalloproteases?

Lines 150-152. Probably the authors should envisage (in a future work) to transfer VIM-2 or SPM-1 to a set of organisms at different evolutionary distances with the (native??) bacteria.

Lines 166-170. The "antitoxic" effect of vesiculation is known for other "membrane toxics"? For instance, in engineered organisms for hyperexpression of proteins of medical or industrial interest. Of course, vesiculation as a membrane-stabilizing process makes sense.

Lines 175-176. There are a number of strains of *E. coli* with VIM-2. How about the possibility of "VIM-2" less-toxic variants in *E. coli*? many VIM-2 derivatives have been found (for instance see Liu Z, Zhang R, Li W, Yang L, Liu D, Wang S, Shen J, Wang Y. 2019. Amino acid changes at the VIM-48 C-terminus result in increased carbapenem resistance, enzyme activity and protein stability. *J Antimicrob Chemother.* 2019 Apr 1; 74(4):885-893. doi: 10.1093/jac/dky536.

Obviously, this is a testable hypothesis looking for more stable *E. coli* strains harboring VIM-2 using in-vitro evolution experiments.

Line 230. Really nice.

Line 256. Again, a nice result.

Line 307. VIM-2 is also found in *Serratia marcescens* strains producing outbreaks.

Line 309. The authors should reconsider some statements about *P. aeruginosa* as the "natural host" of SPM-1. In fact, in the first report about this enzyme, it was suspected (by differences in GC%, I think) that *P. aeruginosa* is NOT the natural host. See Toleman MA, Simm AM, Murphy TA, Gales AC, Biedenbach DJ, Jones RN, Walsh TR. 2002. Molecular characterization of SPM-1, a novel metallo-beta-lactamase isolated in Latin America: report from the SENTRY antimicrobial surveillance programme. *J Antimicrob Chemother.* 2002 Nov; 50(5):673-9.

Lines 320-321. Is there any available phylogeny of NDM-1? "To be tailored" suggest that the enzyme has been evolutionary refined to reduce the costs in a particular host. But what makes NDM-1 to be "uniquely tailored"? Is because it is more ancient than other MBLs, with an extensive history of host-to-host transmission?

Lines 349-350. A phylogeny of signal peptides compared of the phylogeny of hosts could be of interest!

Line 379. How the "amount of produced enzyme" influences fitness? Maybe less transcribed VIM-2 is more palatable for *E. coli* leading to less OMVs)?

Line 392. Do other envelope stresses (as osmolarity) alters OMVs? Do they are additive or synergistic with MBL protein envelope stress?

Line 402. Here comes again the hypothesis of vesicles fusion with otherwise carbapenem-S strains. Is that a real possibility? To my knowledge "resistance protein transfer" has never been described as a mechanism of resistance (!)

Line 409. Here the authors insist in the facilitated transfer of MBL genes. How can be explained? Plasmids entering in vesicles? At least some hypothesis should be presented.

Reviewer #2 (Remarks to the Author):

Antibiotic resistance is a global threat that increasingly curtails our therapeutic arsenal against bacterial infections. While the exposure to antibiotics confers selection to the resistant strains of bacteria, the factors underlying the distribution of resistance cassettes across different hosts under permissive conditions are largely unknown. Lopez et al investigated the evolutionary forces governing the host specificity of different metallo- β -lactamases (MBLs). While the expression of SPM-1 and VIM-2 is well-tolerated by *P. aeruginosa*, both proteins resulted in a significant fitness defect when expressed in *E. coli* and *A. baumannii*. The authors employed a series of biochemical assays to demonstrate that the fitness cost of expressing different MBLs is largely determined by the signal peptide sequence of these proteins. Using *E. coli* as a model, the authors showed that the expression of SPM-1 and VIM-2 triggers an envelope stress leading to hypervesiculation. Swapping the signal peptide of VIM-2 with that of other MBLs endogenous to *E. coli*, alleviated the growth defect associated with its expression. The authors conclude that the signal peptide of SPM-1 and VIM-2 interferes with the efficient translocation of these MBLs across the inner membrane, leading to a growth defect that selects against the dissemination of the MBLs genes in *E. coli* and *A. baumannii*. This is an interesting study that provides new mechanistic insights pertinent to the spread of antibiotic resistance among bacteria. However, I have few major points that require revision:

1-The authors propose that the signal peptide sequences of SPM-1 and VIM-2 are incompatible with the Type I signal peptidases (SPaseI) of *E. coli* and *A. baumannii*, leading to

the inefficient processing of their precursors, and the possible saturation of the Sec translocon. Although the authors used immunoblotting to show the accumulation of VIM-2 and SPM-1 precursors in the cell fractions of *E. coli* and *A. baumannii*, I think they should specifically demonstrate their presence in the inner membrane.

2-The authors proposed in the discussion that *P. aeruginosa* can efficiently translocate SPM-1 and VIM-2 across the inner membrane because it possesses two SPases. Are both SPases required for the translocation of SPM-1/Vim-2? Would the expression of either SPase from *P. aeruginosa* in *E. coli* alleviate the fitness cost associated with expressing SPM-1/Vim-2?

3-The authors relied on protein concentration to quantify OMV in different strains. Given that they demonstrated in figure 1 that the OMV protein cargo differs when MBLs are expressed, using protein concentration as a probe for vesiculation is very confounding. Authors should employ a different method for OMV quantification. Also, how do the authors explain that the OMV protein concentrations were almost identical for the strains expressing NDM-1 and their counterparts carrying the empty vector, though NDM-1 is localized in OMV?

4-Although SPM-1 precursors seem to accumulate more in *A. baumannii* cells compared to VIM-2 (fig 1/S2), the expression of the latter resulted in a longer lag period (Fig 2). The authors should comment on that.

5-I appreciate that the authors included a control for lysis in their OMV purification. Please update fig.1 to show the lysis control.

Minor comments:

- 1-The authors used immunoblotting to quantify DegP levels in MBLs-expressing cells. If densitometry was used for quantification, please report the details in the materials and methods.
- 2-Line 83: "since NDM-containing vesicles can protect bacteria otherwise susceptible to β -lactams, thus enhancing the opportunities for horizontal gene transfer". González et al (2016) showed that OMV carry NDM-1 gene but did not demonstrate HGT using OMV. Please edit that section to properly describe the findings of the cited study.
- 3-Line 172: "The expression of *bla*_{NDM-1} does not enlist a fitness cost...". Please replace "enlist" with "entail".
- 4-Line 183: "These results disclose...". Please replace "disclose".
- 5-Line 204: "A hypervesiculation phenotype can be triggered by two major independent mechanisms". There are other mechanisms for OMV formation that have been proposed (ex. Bonnington KE and Kuehn MJ, 2016; Elhenawy W et al, 2016).
- 6-Line 248: "We previously determined that the cellular localization of NDM-1 and VIM-2 can be altered by swapping the signal peptides of these proteins." Please add the proper citation.
- 7-Please update the materials and methods to describe the statistical analyses conducted in the study.

Reviewer #3 (Remarks to the Author):

I really enjoyed reading the manuscript and looking forward to see published. The paper describe how antimicrobial resistance affects bacterial fitness. Recently I read a paper see J Mol Biol. 2019 Feb 13. pii: S0022-2836(18)31151-3 (maybe would be good to cite in the current manuscript) that discuss the gene and the host's contents, but not answered the specific questions discussed in the current paper; e.g. the effect of specific protein determinants such as the effect of the signal peptide sequence. The work is well executed and I have only one suggestion, would be good to show e.g. SI Figure 6 what is the structure of the thiazolidine L-

VC191.
Juergen Brem

Response to Referees' Letter

Reviewer #1 (Remarks to the Author):

Protein determinants 1 of dissemination and host specificity of metallo- β -lactamases

This extremely interesting and inspiring piece of work indicates that the bacterial host range of MBLs is determined by the impact of MBL protein expression on bacterial fitness. In particular, the authors show the critical role of the protein signal peptide sequence in the production/alleviation of the envelope stress. Membrane vesiculation reduces such stress, encapsulating non-host-adapted MBLs. Many of these findings are really new, and of great potential interest to understand several key-processes in the emergence and epidemiology of antibiotic resistance.

We thank this reviewer very much for his/her positive and insightful comments on the manuscript.

Line 59. "Clinical inhibitors" is not a precise term. I suggest: "Even though there is an active research on this field, inhibitors of these enzymes, largely distributed in Gram negative clinical strains, are not available for therapy".

We appreciate this comment and we have modified this expression (lines 52-54 now) based on this suggestion.

Line 81: *Enterobacter cloacae*

We thank the reviewer for making us note this mistake that has been amended.

Line 83-84. References 17 or 18 does not mention at all horizontal gene transfer. The benefit of being into membranes relates with the ability of functioning with small levels of zinc. It is not clear why NDM-containing vesicles can promote horizontal gene transfer, as NDM genes are not in the vesicle. I can make two hypotheses: i) NDM-vesicles released from the cell with NDM-genes, can fuse with the membranes of antibiotic susceptible cells, and then helping them to resist to carbapenems. That will be cool, but need to be demonstrated. ii) the NDM vesicles can contain plasmids with NDM-genes, and thus these vesicles ALSO promotes transfer. But to my knowledge that also should be demonstrated.

We apologize for not being clear enough. We have now modified this sentence to clarify that OMVs carrying NDM-1 degrade antibiotics, a fact that allows susceptible bacteria to communicate with resistant bacteria. This process would enhance the probability of horizontal gene transfer (HGT) between bacteria by prolonging the lifetime of susceptible bacteria in the presence of antibiotics. Gene transfer mediated by OMVs was first shown by Rumbo and co-workers (Rumbo, C. *et al.* 2011. *Antimicrob. Agents Chemother.* 55, 3084–3090), who demonstrated that OMVs are able to deliver the *bla*_{OXA-24} carbapenem resistance determinant between different *A. baumannii* strains. More recently, Chatterjee *et al.* provided evidence of intra- and inter- species transfer of a plasmid harboring the *bla*_{NDM-1} gene via OMVs (Chatterjee *et al.*, 2017. *J. Antimicrob. Chemother.* 72, 2201–2207). We have modified the text accordingly, including these citations (Lines 76-80).

Line 93. What is the meaning of "non-native bacteria"? The reader might understand that VIM-1 and SPM-1 have evolved into *P. aeruginosa* (or maybe related species). Any support for this assumption?

We agree with the reviewer in that using "non-native" or "native" bacterial host can be misleading. By "native" or "non-native bacterial hosts" we referred to more or less frequent microbial hosts producing VIM-2 or SPM-1 according to epidemiological reports (Hong, D. J. *et al.* 2015. *Infect. Chemother.* 47, 81; Toleman, M. A. *et al.* 2002. *J. Antimicrob. Chemother.* 50, 673–9; Zhao WH· Hu ZQ. 2011. *Future Microbiol.* 6:3, 317-333).

We have now replaced these terms by "non-frequent" bacterial host and "frequent" bacterial host.

Lines 100-102. Is that also the case for other metalloproteases?

NDM enzymes are the only family of metallo- β -lactamases with a lipobox sequence in the signal peptide sequence. We have not found evidences of other metalloproteases (devoid of lactamase activity) with lipobox sequences in bacteria.

Lines 150-152. Probably the authors should envisage (in a future work) to transfer VIM-2 or SPM-1 to a set of organisms at different evolutionary distances with the (native??) bacteria.

We thank the reviewer for his/her interesting suggestion. In this work we focused on bacterial models of Gram-negative pathogens that have emerged as producers of MBLs and have been highlighted as priority 1 by the WHO (entity/mediacentre/news/releases/2017/bacteria-antibiotics-needed/en/index.html). We are indeed planning in the near future to work on another set of distantly related bacteria aimed to better understand the molecular determinants defining the bacterial host range of MBLs.

Lines 166-170. The "antitoxic" effect of vesiculation is known for other "membrane toxics"? For instance, in engineered organisms for hyperexpression of proteins of medical or industrial interest. Of course, vesiculation as a membrane-stabilizing process makes sense.

The field of study of OMVs is very broad, and also includes cases of engineered organisms. Here we focus on cases in which a direct relationship between stress and protein-induced toxicity has been addressed at the molecular level. The role of OMV release as an envelope stress response was shown in several previous works (McBroom AJ, Kuehn MJ. 2007. *Mol Microbiol.*, 63:545-558; Anand, D, Chaudhuri, A. 2016. *Molecular Membrane Biology*, 33:6-8, 125-13). In the case of membrane acting-stressors or membrane-perturbing substances, it has been observed that the treatment of *P. aeruginosa* cells with polymyxin B or D-cycloserine (both compounds that perturb the outer membrane) has resulted in an increased production of vesicles (Macdonald IA, Kuehn MJ. 2013. *J Bacteriol.* 195,13:2971-81).

Lines 175-176. There are a number of strains of *E. coli* with VIM-2. How about the possibility of "VIM-2" less-toxic variants in *E. coli*? many VIM-2 derivatives have been found (for instance see Liu Z, Zhang R, Li W, Yang L, Liu D, Wang S, Shen J, Wang Y. 2019. Amino acid changes at the VIM-48 C-terminus result in increased carbapenem resistance, enzyme activity and protein stability. *J Antimicrob Chemother.* 2019 Apr 1;74(4):885-893. doi: 10.1093/jac/dky536. Obviously, this is a

testable hypothesis for more stable *E. coli* strains harboring VIM-2 using in-vitro evolution experiments.

The number of members of *Enterobacteriaceae* family producing VIM-2 or its variants is very limited (Zhao WH, Hu ZQ. 2011. *Future Microbiol.* 6:3, 317-333; <https://www.ncbi.nlm.nih.gov/pathogens/isolates#/search/vim>). In fact, we found only one report that identified an *E. coli* clinical isolate producing VIM-2 (Galani *et. al.* 2004. *Clin. Microbiol. Infect.*, 10: 757-760). It is likely that this strain could have some mutations favoring expression of VIM-2 with a reduced fitness cost. Unfortunately, sequence information on this strain is not available in the paper. This was included in the discussion section (lines 320-322).

The reviewer mentions the possibility that there are less toxic variants of VIM-2 that can be expressed and maintained in *E. coli* strains without generating a fitness cost. This is likely, and may deserve a further study in the future. In the particular case of the VIM-48 variant mentioned by the reviewer, the enzyme was isolated from a *Pseudomonas putida* strain DZ-C20 and up to date there are no reports of VIM-48 producing *E. coli* isolates (<https://www.ncbi.nlm.nih.gov/protein/?term=vim-48>).

We are planning in the near future *in vitro* evolution experiments to find more stable, adapted and versatile *E. coli* strains for the expression of VIM-2 or SPM-1.

Line 230. Really nice.

Thanks for your comment.

Line 256. Again, a nice result.

Thanks for your comment.

Line 307. VIM-2 is also found in *Serratia marcescens* strains producing outbreaks.

We thank the reviewer and we apologize for this omission. We have now included this statement in the lines 320-322.

Line 309. The authors should reconsider some statements about *P. aeruginosa* as the "natural host" of SPM-1. In fact, in the first report about this enzyme, it was suspected (by differences in GC%, I think) that *P. aeruginosa* is NOT the natural host. See Toleman MA, Simm AM, Murphy TA, Gales AC, Biedenbach DJ, Jones RN, Walsh TR. 2002. Molecular characterization of SPM-1, a novel metallo-beta-lactamase isolated in Latin America: report from the SENTRY antimicrobial surveillance programme. *J Antimicrob Chemother.* 2002 Nov; 50(5):673-9.

We agree with the reviewer and we decided change of statements where appears "native host" or "natural host" for "frequent host" or "usual host", since we want to highlight that VIM-2 and SPM-1 are mostly disseminated in clinical isolates of *Pseudomonas aeruginosa* resistant to carbapenems, not focusing on the original source of *bla* genes, which in most cases is unknown.

Lines 320-321. Is there any available phylogeny of NDM-1? "To be tailored" suggest that the enzyme has been evolutionary refined to reduce the costs in a particular host. But what makes NDM-1 to be "uniquely tailored"? Is because it is more ancient than other MBLs, with an extensive history of host-to-host transmission?

The expression “uniquely tailored” refers to the presence of the lipobox, that is present only within the NDM family, and absent in the rest of clinically relevant MBLs. This molecular feature enables selective secretion into OMVs, a crucial result for our work, and we wanted to highlight it in this way. It is clear that the whole presence of a lipobox is evolutionary most costly than a point mutation. The phylogeny of NDMs shows that this lipobox is preserved in all members of this family (Zhihai L et al. 2018. *Front Microbiol.* 9:248).

Lines 349-350. A phylogeny of signal peptides compared of the phylogeny of hosts could be of interest!

This is a very interesting point, and we thank the reviewer for this suggestion. We have now performed sequence alignments and phylogenetic trees of signal peptides and mature proteins from the B1 MBLs for which there is information regarding the bacterial hosts. As models of broad host range enzymes we used IMP-1, NDM-1, GIM-1, DIM-1 and VIM-1 (orange color in the figure), which were reported in members of *Enterobacteriaceae* and *Pseudomonadales* (<https://www.ncbi.nlm.nih.gov/pathogens/isolates#>, <https://www.ncbi.nlm.nih.gov/protein/>). As model of narrow bacterial host range enzymes we chose VIM-2, SPM-1, SIM-1 and IMP-31 (blue color in the figure), which were almost exclusively identified in *Pseudomonadales*.

This phylogeny does not reveal any clustering of the signal peptides according to the enzymes' host range, indicating that there are not obvious sequence determinants regulating MBL distribution among bacterial species. Instead, we observed a similar phylogenetic distribution for signal peptides and the corresponding mature proteins, indicating that signal peptides are not independent modules but instead co-evolve with the rest of the protein sequences, as clearly seen in the case of VIM-1 and VIM-2 (or IMP-1 and IMP-31) exhibiting different bacterial host specificities.

We then analyzed the particular cases of hybrid enzymes within the VIM family. Similar to VIM-12, we found another hybrid enzyme, VIM-25, which contains the signal peptide of VIM-1 and the mature protein sequence of VIM-2. We wondered whether the signal peptides are acting as independent modules regulating the host range of the enzymes in these cases. However, as the phylogenetic tree shows, the mature VIM-12 and VIM-25 are actually hybrids between VIM-2 and VIM-1 families, suggesting again that the signal peptide has co-evolved with the mature protein sequence.

Overall, considering that MBLs are highly divergent enzymes, we propose that signal peptides followed different particular evolutionary pathways without converging into a specific sequence pattern.

We have now added a comment on this issue and added a Figure with the phylogeny (lines 278-280 and Supplementary Figure 8).

Line 379. How the "amount of produced enzyme" influences fitness? Maybe less transcribed VIM-2 is more palatable for *E. coli* leading to less OMVs?

The reviewer is correct. This is a direct conclusion from our experiments. As discussed in lines 187-190 and shown in Figure S6, larger concentrations of IPTG induce higher expression levels of VIM-2 and SPM-1 that further retard the growth of *E. coli* and *A. baumannii*, so it is evident that these bacterial hosts are not adapted to express these enzymes at large quantities, unlike *P. aeruginosa* strains.

Line 392. Do other envelope stresses (as osmolarity) alters OMVs? Do they are additive or synergistic with MBL protein envelope stress?

There are several reports linking cell envelope stress with increased OMVs productions, such as accumulation of misfolded proteins, temperature, chelators of outer membrane divalent cations, etc. (Anand, D, Chaudhuri, A. 2016. *Molecular Membrane Biology*, 33:6-8, 125-13). In the particular case of osmotic stress, it has been shown that cells of *P. putida* released more vesicles when exposed to NaCl (Eberlein, Christian *et al.* 2018. *Applied microbiology and biotechnology* 102,6: 2583-2593). We have not assessed the effect of combining different envelope stresses with MBL protein envelope stress since we wanted to identify the molecular determinants of host adaptability. This is a great suggestion for future work.

Line 402. Here comes again the hypothesis of vesicles fusion with otherwise carbapenem-S strains. Is that a real possibility? To my knowledge "resistance protein transfer" has never been described as a mechanism of resistance (!)

We have previously shown that *E. coli* cells susceptible to β -lactam antibiotics became transiently resistant when incubated with NDM-1-containing vesicles (González, L. J. *et al.* 2016. *Nat. Chem. Biol.* 12, 516–522) and not by horizontal gene transfer. Interestingly, we detected the presence of NDM-1 protein from the vesicles in the susceptible bacteria (González, L. J. *et al.* 2016. *Nat. Chem. Biol.* 12, 516–522), suggesting that at least part of the protection came from vesicle association and/or fusion to receptor cells. In fact, there are many reports demonstrating the ability of vesicles to fuse to receptor membranes (other bacterial cells or eukaryotic cells) (O'Donoghue EJ and Anne Marie Krachler AM. 2016. *Cell Microbiol.* 18, 11: 1508-1517; Bitto N *et al.* 2017. *Sci Rep.* 7(1):7072) as well as resistance protein transfer mediated by OMVs (Schaar V *et al.* 2011. *Antimicrob Agents Chemother.* 55(8):3845-53; Schaar V *et al.* 2013. *J Antimicrob Chemother.* 68: 593–600).

It is important to note that the protection to carbapenem-susceptible bacteria does not necessarily require the fusion of OMVs with the acceptor bacteria, since free surrounding OMVs containing NDM-1 can hydrolyze beta-lactam antibiotics efficiently.

Line 409. Here the authors insist in the facilitated transfer of MBL genes. How can be explained? Plasmids entering in vesicles? At least some hypothesis should be presented.

This important issue was answered more clearly above (in the point regarding lines 83-84). Moreover, in the lines 419-425 we commented that OMVs-mediated *bla*_{NDM-1} gene transfer has been demonstrated in previous works and we cited the study of Chatterjee *et al.*, who have showed the presence of the plasmid containing *bla*_{NDM-1} into OMVs from *A. baumannii*.

Reviewer #2 (Remarks to the Author):

Antibiotic resistance is a global threat that increasingly curtails our therapeutic arsenal against bacterial infections. While the exposure to antibiotics confers selection to the resistant strains of bacteria, the factors underlying the distribution of resistance cassettes across different hosts under permissive conditions are largely unknown. López et al investigated the evolutionary forces governing the host specificity of different metallo- β -lactamases (MBLs). While the expression of SPM-1 and VIM-2 is well-tolerated by *P. aeruginosa*, both proteins resulted in a significant fitness defect when expressed in *E. coli* and *A. baumannii*. The authors employed a series of biochemical assays to demonstrate that the fitness cost of expressing different MBLs is largely determined by the signal peptide sequence of these proteins. Using *E. coli* as a model, the authors showed that the expression of SPM-1 and VIM-2 triggers an envelope stress leading to hypervesiculation. Swapping the signal peptide of VIM-2 with that of other MBLs endogenous to *E. coli*, alleviated the growth defect associated with its expression. The authors conclude that the signal peptide of SPM-1 and VIM-2 interferes with the efficient translocation of these MBLs across the inner membrane, leading to a growth defect that selects against the dissemination of the MBLs genes in *E. coli* and *A. baumannii*. This is an interesting study that provides new mechanistic insights pertinent to the spread of antibiotic resistance among bacteria. However, I have few major points that require revision:

We thank this reviewer for his/her positive comments on our work.

1-The authors propose that the signal peptide sequences of SPM-1 and VIM-2 are incompatible with the Type I signal peptidases (SPaseI) of *E. coli* and *A. baumannii*, leading to the inefficient processing of their precursors, and the possible saturation of the Sec translocon. Although the authors used immunoblotting to show the accumulation of VIM-2 and SPM-1 precursors in the cell fractions of *E. coli* and *A. baumannii*, I think they should specifically demonstrate their presence in the inner membrane.

This is an excellent suggestion. We agree with the reviewer and we have taken into account this important issue by performing new experiments. We now provide evidence that the precursor proteins of VIM-2 and SPM-1 accumulate in the periplasmic face of the inner membrane from *E. coli* cells expressing these MBLs.

We fractionated *E. coli* cells expressing VIM-2 and SPM-1 into periplasm (P) and spheroplasts (S_T) by EDTA-Lysozyme treatment (González, L. J. *et al.* 2016. *Nat. Chem. Biol.* 12, 516–522). Spheroplasts were then lysed and separated into soluble (S_S) and insoluble (S_I: insoluble) fractions by centrifugation. The mature and precursor forms of the MBLs could be identified in the different fractions by immunoblot. While the precursor forms were the predominant species in the insoluble fraction (S_I), both for VIM-2 and SPM-1, the periplasmic fraction contained only the mature forms of these proteins. We observed a minor proportion of mature proteins in the spheroplasts, more notable in SPM-1, which would correspond to the newly processed enzymes in a previous step before release into the soluble fraction of the periplasm.

Finally, we performed proteolysis experiments with Proteinase K in intact spheroplasts (Sph) to assess whether the insoluble precursor proteins are located in the periplasmic or cytoplasmic face of the inner membrane. As Figure 1b shows, precursor forms of VIM-2 and SPM-1 were accessible to digestion by the protease, while the cytoplasmic protein GroEL remained intact, indicating that precursor forms of VIM-2 and SPM-1 are bound to the periplasmic face of the inner membrane.

We have now included this experiment as Figure 1b and Supplementary Figure 2, that is discussed in the main text (lines 131-134), with the corresponding experimental part (lines 550-557, lines 949-950).

2-The authors proposed in the discussion that *P. aeruginosa* can efficiently translocate SPM-1 and VIM-2 across the inner membrane because it possesses two SPases. Are both SPases required for the translocation of SPM-1/Vim-2? Would the expression of either SPase from *P. aeruginosa* in *E. coli* alleviate the fitness cost associated with expressing SPM-1/Vim-2?

This is an excellent question that we are unable to address at this point, and we realize we have been over speculative in the discussion regarding this aspect. Accordingly, we have toned down our proposal based on this suggestion (lines 378-381).

3-The authors relied on protein concentration to quantify OMV in different strains. Given that they demonstrated in figure 1 that the OMV protein cargo differs when MBLs are expressed, using protein concentration as a probe for vesiculation is very confounding. Authors should employ a different method for OMV quantification. Also, how do the authors explain that the OMV protein concentrations were almost identical for the strains expressing NDM-1 and their counterparts carrying the empty vector, though NDM-1 is localized in OMV?

We thank the reviewer for noting this. The method of OMV quantification used in this work, which measures total protein content in vesicles, is the most commonly employed by experts from the field of vesicles (Chutkan, H. et al. 2013. *Methods Mol. Biol.* 966:259-72). OMV quantification by protein dosage is governed by outer membrane proteins (OMPs), which are the major proteins present in OMVs (Schwechheimer, C. et al. 2013. *Biochemistry* 52, 3031–40). Given that the levels of MBLs incorporated into vesicles were insignificant compared to the major outer membrane proteins (OMPs); we consider that the methodology of OMV quantification used in this work is adequate. This can be clearly seen in the SDS-PAGEs of OMVs shown in Figure Supplementary S5. We have now clarified this point in lines 147-148 and 578-581.

4-Although SPM-1 precursors seem to accumulate more in *A. baumannii* cells compared to VIM-2 (fig 1/S2), the expression of the latter resulted in a longer lag period (Fig 2). The Authors should comment on that.

The reviewer raises an important issue. We apologize for not having been clear enough. The impact of VIM-2 expression on *A. baumannii* growth is more marked than when expressing SPM-1, which is evidenced by a prolonged lag period. It is likely that the higher toxicity of VIM-2 compromises cell metabolism and thus protein synthesis, resulting in lower levels of precursor and mature VIM-2, as observed in Figure 1a. However, in stationary phase, where the proteins were able to accumulate for a longer period, we detected higher amounts of VIM-2, evidencing the accumulation of the precursor form (Fig. S3). We have now clarified this aspect on the main text (in lines 134-139 and 183-186).

5-I appreciate that the authors included a control for lysis in their OMV purification. Please update fig.1 to show the lysis control.

We thank the reviewer for pointing out this important issue. We had performed this lysis control by cytoplasmic GroEL detection. We have now completed Figure 1a with immunoblot of GroEL and incorporated this information in the legend of Figure 1a (lines 947-948).

Minor comments:

1-The authors used immunoblotting to quantify DegP levels in MBLs-expressing cells. If densitometry was used for quantification, please report the details in the materials and methods.

Modified as indicated.

2-Line 83: "since NDM-containing vesicles can protect bacteria otherwise susceptible to β -lactams, thus enhancing the opportunities for horizontal gene transfer". González et al (2016) showed that OMV carry NDM-1 gene but did not demonstrate HGT using OMV. Please edit that section to properly describe the findings of the cited study.

We apologize for not being clear enough. We have now modified this sentence to clarify that OMVs carrying NDM-1 degrade antibiotics, a fact that allows susceptible bacteria to communicate with resistant bacteria. This process would enhance the probability of horizontal gene transfer (HGT) between bacteria by prolonging the lifetime of susceptible bacteria in the presence of antibiotics. Gene transfer mediated by OMVs was first shown by Rumbo and co-workers (Rumbo, C. et al. 2011. *Antimicrob. Agents Chemother.* 55, 3084–3090), who demonstrated that OMVs are able to deliver

the *bla*_{OXA-24} carbapenem resistance determinant between different *A. baumannii* strains. More recently, Chatterjee *et al.* provided evidence of intra- and inter- species transfer of a plasmid harboring the *bla*_{NDM-1} gene via OMVs (Chatterjee *et al.*, 2017. *J. Antimicrob. Chemother.* 72, 2201–2207). We have modified the text accordingly, including these citations (lines 76-80).

3-Line 172: "The expression of *bla*_{NDM-1} does not enlist a fitness cost...". Please replace "enlist" with "entail".

Modified as suggested.

4-Line 183: "These results disclose....". Please replace "disclose".

Changed as indicated.

5-Line 204: "A hypervesiculation phenotype can be triggered by two major independent mechanisms". There are other mechanisms for OMV formation that have been proposed (ex. Bonnington KE and Kuehn MJ, 2016; Elhenawy W et al, 2016).

We thank the reviewer for underlining this point. We have now modified this sentence accordingly (lines 214-215).

6-Line 248: "We previously determined that the cellular localization of NDM-1 and VIM-2 can be altered by swapping the signal peptides of these proteins." Please add the proper citation.

The citation was added.

7-Please update the materials and methods to describe the statistical analyses conducted in the study.

We have included the standard deviations for sample replicates, as requested. These were included in the figure legends.

Reviewer #3 (Remarks to the Author):

I really enjoyed reading the manuscript and looking forward to see published. The paper describe how antimicrobial resistance affects bacterial fitness. Recently I read a paper see J Mol Biol. 2019 Feb 13. pii: S0022-2836(18)31151-3 (maybe would be good to cite in the current manuscript) that discuss the gene and the host's contents, but not answered the specific questions discussed in the current paper; e.g. the effect of specific protein determinants such as the effect of the signal peptide sequence. The work is well executed and I have only one suggestion, would be good to show e.g. SI Figure 6 what is the structure of the thiazolidine L-VC191.

Juergen Brem

Thank the reviewer for pointing out the citation; we have now included it (lines 122-125).

Regarding thiazolidine L-VC191, we have repeated the experiment using an already reported inhibitor, *i.e.*, bithiazolidine L-CS319.

REVIEWERS' COMMENTS:

Reviewer #2 (Remarks to the Author):

Thank you for addressing my comments!
Wael Elhenawy, PhD

Response to Referees' Letter

Reviewer #2 (Remarks to the Author):
Thank you for addressing my comments!
Wael Elhenawy, PhD

We thank the reviewer for the comments.